# Nurses' knowledge, perceived challenges, and recommended solutions regarding premature infant care: A mixed method study in the referral and tertiary hospitals in Dar es salaam, Tanzania

Mwajuma Mwikali[1,2‡]*, Nahya Salim[1‡]*, Isabella Sylvester[3], Emmanuel Munubhi[1]

1 Department of Peadiatrics and Child Health, School of Medicine, Muhimbili University of Health and Allied Sciences (MUHAS), Dar es Salaam, Tanzania, 2 Department of Peadiatrics and Child Health, Msambweni County Referral Hospital (MCRH), Kwale, Kenya, 3 Department of Peadiatrics and Child Health, Muhimbili National Hospital, Dar es Salaam, Tanzania

‡ MM and NS are joint first authors on this work.
* Bejummer@gmail.com (MM); nsalim@ihi.or.tz (NS)

## Abstract

### Background

There has been an increase in preterm birth of about 2% in a span of 14 years (2000–2014) mainly from Asia and Sub-Saharan Africa. Nursing care is very crucial and a lack of knowledge of health care providers is a contributing factor to morbidity and mortality. With the increasing number and investment of preterm infants towards attaining sustainable development goals (SDG) 3.2, nurses' knowledge adequacy, challenges and solutions on their care needs to be affirmed.

### Methods

A mixed method study was conducted between September 2020 to January 2021 in the neonatal units of four hospitals in Dar es Salaam. Self-administered structured questionnaire was used to assess adequacy of knowledge set at 50% or more for the three main domains 1) Essential newborn Care 2) Infection prevention and management 3) Special care and monitoring. A phenomenological design using a structured interview guide focused on challenges and recommended solutions in acquiring on-the- job training on the care of preterm infants. Quantitative data were analyzed using SPSS version 23 and qualitative data were thematically categorized.

### Results

Out of 52 of nurses who participated and providing care to preterm infants; 48.1% came from a tertiary hospital, (84.6%) were females, only 28.8% aged more than 40 years and 23.1% had less than one year of experience. Overall, 55.8% of the nurses had never received on job training. Adequate knowledge among nurses was 94% on essential

**Data Availability Statement:** All relevant data are within the paper and its supporting information files.

**Funding:** The authors received no specific funding for this work.

**Competing interests:** The author have declared that no competing interest exist.

newborn care, 80.8% on infection prevention and management and 36.5% on special care and monitoring of preterm infants. Generally, immediate actions of helping baby breath (HBB) and cord care scored poorest. Components on special care and monitoring which had lowest scores included blood glucose monitoring, temperature monitoring and acceptable daily weight gain. Being more than 41 years old, a female nurse and working in the neonatal unit for at least 1–3 years were more likely to determine adequacy of knowledge on infection prevention and management. Lack of schedule and ways to identify those who require training were among the challenges mentioned in the focus group discussion.

## Conclusion

The findings demonstrate an urgent need of instilling knowledge, skills and competences among nurses providing preterm care in our hospitals. Most nurses had not attended training on the care of premature infant. Special care and monitoring were most poorly performed. The recommended solutions included continuous medical education (CME) for all nurses through hospital and government commitment and encourage mentorship within and between hospitals. Nurses who are female, older than 41 years and those with 1 to 3 years of experience should be considered when planning for CME and mentorship program on infection prevention and management.

## Introduction

Neonatal deaths are increasingly contributing to under-five deaths (from 40% in 1990 to 47% in 2017 respectively). Sub-Sahara Africa has the highest neonatal mortality rate (NMR) at 28 per 1,000 live births compared to other settings contributing to almost half of under-five mortality [1]. Of the 2.5 million neonatal deaths that occur globally, 75% is contributed by prematurity and its complications. Most of these deaths are potentially preventable through improving availability of skilled providers, quality of antenatal and postnatal care and care of small and sick newborn [2]. The World Health Organization (WHO) defines a premature infant as babies born alive before 37 weeks of pregnancy are completed. Due to prematurity, these infants are prone to short- and long-term complications requiring attention [3]. In low-resource settings one of the factors that contribute to neonatal mortality is a human resource gap in terms of inadequate knowledge, low staff competence and health worker shortage contributing to inadequate care provided to these infants [4, 5].

There has been an increase in preterm birth rate globally but more so in Sub-Saharan Africa including Tanzania [6]. Increase in premature infants means an increased need to invest on quality care of preterm babies as global review of evidence suggest [4]. With 77 to 83% of birth happening in facility, there is an urgent need to invest and improve quality of small and sick newborn care (SSNC) in Tanzania [7]. The government of Tanzania is committed to attaining sustainable development goal (SDG) 3.2 by 2030, targeting to reach 15 neonatal deaths per 1000 live births by 2025 [7]. Despite nurses being the backbone care providers of premature infants, most of the available studies have assessed nurses' knowledge, neither challenges nor solutions have been identified. Understanding the level of knowledge in caring of preterm infants and the factors affecting them will be a step closer to improving care considering the burden of prematurity in terms of both morbidity and mortality. This study assessed nurses' knowledge on the care of premature babies and their experiences including challenges and recommended solutions in acquiring the knowledge.

## Methodology

A mixed method was used combining both descriptive cross-sectional study with qualitative phenomenological study design. The quantitative component determined the nurses' knowledge and associated factors whereas the qualitative component was an additive to the study to determine general experience, challenges, and proposed solutions regarding the acquisition of knowledge about premature infant care among the neonatal unit nurses. The study was conducted in the Dar es Salaam region, the largest city in Tanzania with a population of 5.5 million [8]. Dar es Salaam has five districts including kinondoni, Ilala, Temeke, Ubungo, and Kigamboni. There are three regional referral hospitals (RRH) namely Amana, Mwananyamala and Temeke. All the three RRHs refer newborns to two tertiary hospitals, Muhimbili National Hospital (MNH), Upanga, and MNH-Mloganzila. This study was conducted in the three RRHs and one of the tertiary hospitals, MNH- Upanga. MNH-Upanga alone accounts for approximately 369 preterm deliveries per month excluding those referred in from other hospitals. Bed capacity being approximately 100 in the specific premature ward. In the regional referral hospitals, preterm deliveries can be approximately 42 per month in all the hospitals with a bed capacity of approximately 15 to 20 which is for both term and preterm newborns. It should be noted that the hospital capacity provided was prior to investment implemented by the Newborn Essential Solutions and Technologies (NEST 360) program in collaboration with the ministry of health in Tanzania. Premature newborns weighing less than 1.8kg and those that are sick irrespective of their weight are admitted to the neonatal unit. Each hospitals had a neonatal intensive care unit (NICU)/ corner that offered special service and care for premature infants. Since all hospitals provided newborn care, it gave room for comparability in terms of the level of knowledge.

Because of limited number of staff, study population were all nurses who worked in the neonatal unit in these hospitals and those that specifically took part in caring for premature infants. Nursing cadre that takes part in the care include enrolled nurse (EN), assistant nursing officer (ANO), and nursing officer (NO). The study was conducted between September 2020 to January 2021.

Based on the study done in Uganda (6), the proportion of nurses that had adequate knowledge in identifying and caring for low birth weigh infants was 69%. Using this proportion, with a margin of error ($\varepsilon$) of 0.05 in the Kish Leslie formula [9]:

$\varepsilon$ = margin of error

Z = z-score

P = proportion

n0 = sample size

$$n0 = \frac{Z^2 P\ (1-P)}{\varepsilon^2}$$

$$\frac{1.96^2 x 0.69\ (1-0.69)}{0.05^2} = 328$$

Then the formula for a finite population correction for proportion to acquire a feasible sample size was used [9]. The total population of nurses in MNH, Mwananyamala, Amana, Temeke were approximately 26, 9, 9,10 respectively. Making it a total of 54.

n = adjusted sample size

n0 = sample size calculated = 328

N = population size = 54

$$n = \frac{n_0}{1 + \frac{(n_0 - 1)}{N}}$$

n = 328/ (1+ (328-1/54))

n = 46

Based on the proportion of the population, recruited nurses from MNH were 25 (48%), Temeke 10 (19%), Mwananyamala 8 (15%), Amana 9 (17%).

Those excluded from the study were nurses who declined consent to participate. Data was collected using a self-administered questionnaire (S1 File) which was in both English and Swahili version with the tool adopted from a study done in Masindi Uganda [10] after seeking permission to adapt the tool. The tool was later modified using the National guideline for neonatal care and establishment of neonatal care units of August, 2019 [11]. A pretest was done with 5% of the required sample at the Muhimbili National Hospital, among neonatal nurses who worked in the term infants. Some questions were vague and some answers in the multiple choices didn't make sense to the nurses. Based on the pilot findings, the questionnaire was modified accordingly and used in the data collection for the study. Questionnaires were distributed to assigned trained nurses in each regional hospital that assisted in data collection. At MNH—Upanga data was collected by the principal investigator, student resident responsible for the study.

Consecutive sampling was done until sample size was attained in all hospitals.

Quantitative data was entered and cleaned in SPSS Version 23. No missing data was found. The dependent variables were knowledge on three main domains: 1. Essential newborn care 2. Infection prevention and management and 3. Special care and monitoring. The independent variables were sociodemographic factors (age, gender, cadre), institutional factors (either working in the regional hospital or the national hospital), level of education, years of experience, and when was the last training/workshops on preterm care attended. Frequency tables were generated to determine the social demographic distribution of the nurses. Ranges were generated in the variables of age and years of experience.

The questions were computed, and each question was graded as 1 and 0 and dichotomized as correct and incorrect responses respectively. For each of the themes, nurses were judged to have adequate knowledge if they mentioned correctly two or more of the essential newborn care questions, two or more of infection prevention and management, and seven or more questions answered correctly in special care and monitoring section which was equivalent to more than 50% in each domain [10].

Chi-square or Fischer's test p-value was used for the categorical variables against adequate and inadequate knowledge in each domain to determine association factors and any p-value of < 0.2 in any variable which was significant was entered into binary logistic regression. Upon doing the bivariate analysis, a p-value of <0.2 in any of the variables and those which are of clinical importance underwent further multivariate analysis. A p-value of < 0.05 was determined as statistically significant in the association of nurses' knowledge in the care of preterm newborns. The qualitative study was conducted by the female corresponding author, MM who at the time was a resident in Paediatrics and child health. MM had undergone basic training in qualitative study design when doing her training. Qualitative data have been reported following the checklist for COnsolidated criteria for REporting Qualitative research [12]. Some of the nurses at Muhimbili National Hospital had worked with the MM while doing her neonatal rotation. Those in the regional referral hospital she had met them when collecting quantitative data. No participant had reported any character of the interviewer. Phenomenological theory was used for focus group discussion and purposive sampling of nurses who worked in newborn unit caring for preterm infants. Face to face approach was used. A total of 10 participants were included in the study. Whereby in each focus group there were 5 nurses. This was decided based on previous study done in India [13]. The focus group discussion was done in the newborn unit teaching rooms in each respective hospital. While conducting the FGD, there was only the researcher and participants present.

The majority were female nurses (90%). The participant's age ranged from 29 to 56 years with the mean age of 42.4 years. The nurse who had worked in the neonatal unit the longest had 17 years of experience while those who had worked the shortest time had one year of experience. Out of the total, enrolled nurses were 4, nursing officer were 3 and assistant nursing officer were 3. Six of the participants had at least attended the training of preterm care within the last five years while working in the neonatal unit. Nurses from Temeke RRH were chosen to represent the three regional hospitals based on the overall knowledge adequacy and environmental similarities. An interview guide (S2 File) with questions was used and prompting was done where answering was insufficient. The first interview was done in the regional referral hospital with the most poorly responded questions during quantitative data collection. This was aimed at answering one of the objects of challenges and recommended solutions in acquiring on-the- job training.

The second FDG was conducted in the national hospital to get a variety of responses both in the national and regional referral hospital. Saturation was attained once the objective questions were answered.

Interviews were recorded in an audio application installed in the phone. Notes were taken as well during the discussion where probing was done. FGD in the regional hospital took thirteen minutes despite probing of participants. While in the national hospital it took twenty-five minutes where one participant was more engaging and interactive. Transcripts were not returned to the participants for clarification.

There were 2 data coders. Recordings were transcribed and with the assistance of an expert in qualitative data analysis, the transcribed notes were read several times to determine codes using the codebook created. The codes were written in MS Excel then made into categories, subthemes, and themes. Themes derived included general experience in acquiring knowledge, challenges perceived, and solutions to perceived challenges.

## Ethical clearance and considerations

In line with ethical issues, ethical clearance was sorted from the Muhimbili University of Health and Allied Sciences Research and Publication committee prior to the study with the ethical approval number MUHAS-REC-06-2020-307. Then permission was attained from each hospital research directorate and thereafter from the consecutive sampling of the nurses' neonatal unit of each hospital that cares for the premature babies. Participants were informed of the research and the main objectives, and a written informed consent was obtained. Autonomy, voluntary participation, and withdrawal from the study at any time during the research were stated and determined throughout data collection. Confidentiality was maintained during the entire course of the study by using serial numbers during the administration of the questionnaires and only the principal investigator had access to the data. After data collection and analysis, it was obvious that less than 50% of the nurses from the regional referral hospitals knew how to resuscitate using the Helping Baby Breath (HBB) protocol. This was communicated to the in charges of the neonatal ward in the respective hospitals to ensure continuous medical education was conducted.

## Results

### Quantitative results

A total of 52 nurses were enrolled in which 9 (17%) were from Amana, Mwananyamala 8 (15%), Temeke 10 (19%), and the MNH, Upanga 25 (48.1%). Almost 85% (44/52) of the nurses were female. At the time of the study, 56% (29/52) of the nurses had never had the privilege to attend training on the care of premature infants (Fig 1).

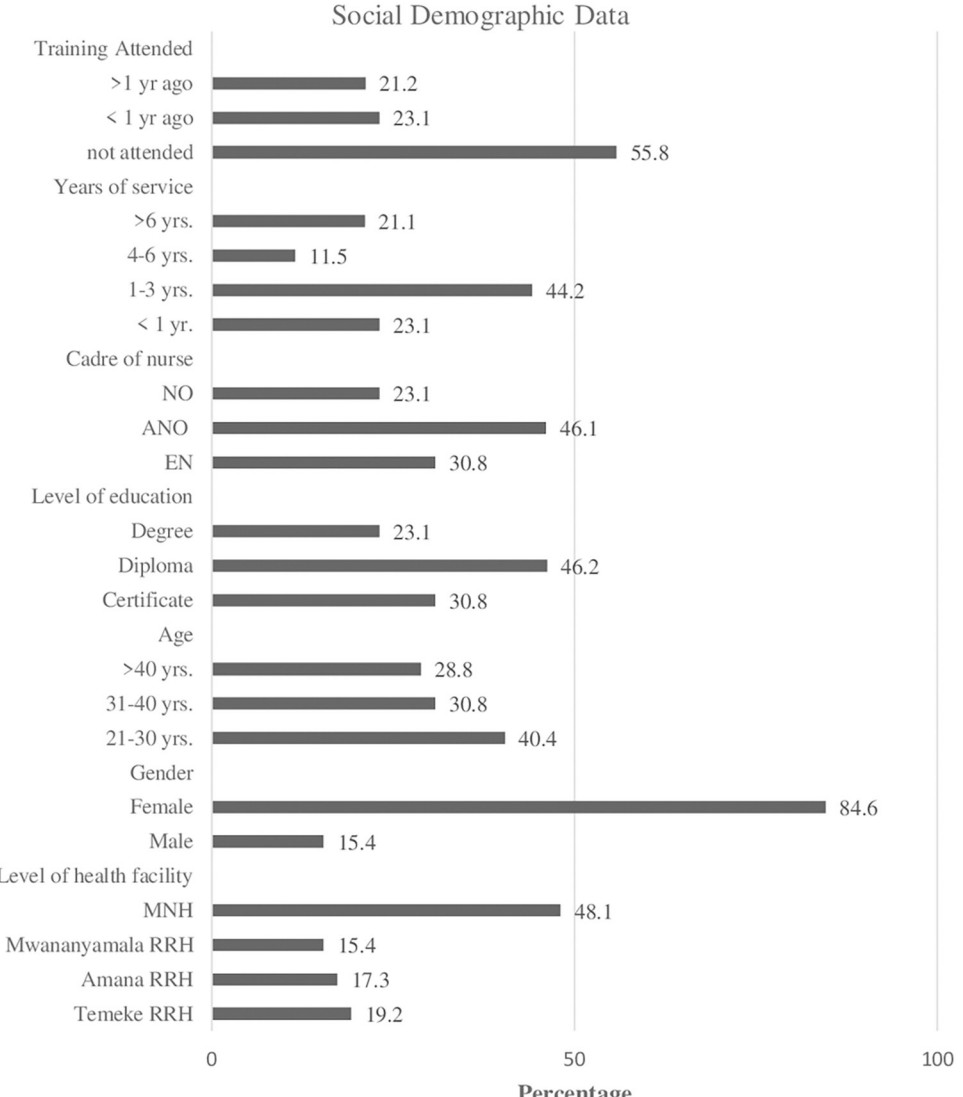

**Fig 1. Social demographic characteristics.**

In essential newborn care, overall, all hospitals had a poor response on HBB question compared to other questions in the same domain. In infection prevention and management, overall umbilical cord care response was poor compared to other questions in this domain (Fig 2).

Knowledge on special care and monitoring provided to the premature infants was poorly responded in the following questions: 19% knew how to do blood glucose monitoring within the first 24 hours after delivery, temperature monitoring in the preterm newborn with less than 1500 g babies with 25% of the nurses having a correct response and only 29% of the nurses knew the acceptable daily weight gain (Fig 3).

**Factors associated with nurses' knowledge.** In the essential newborn care to note is that those who were aged 41 years and above, 15/15 (100%) had adequate knowledge but there was no association of knowledge on essential newborn care with the different age groups. (p = 0.77) as shown in Table 1.

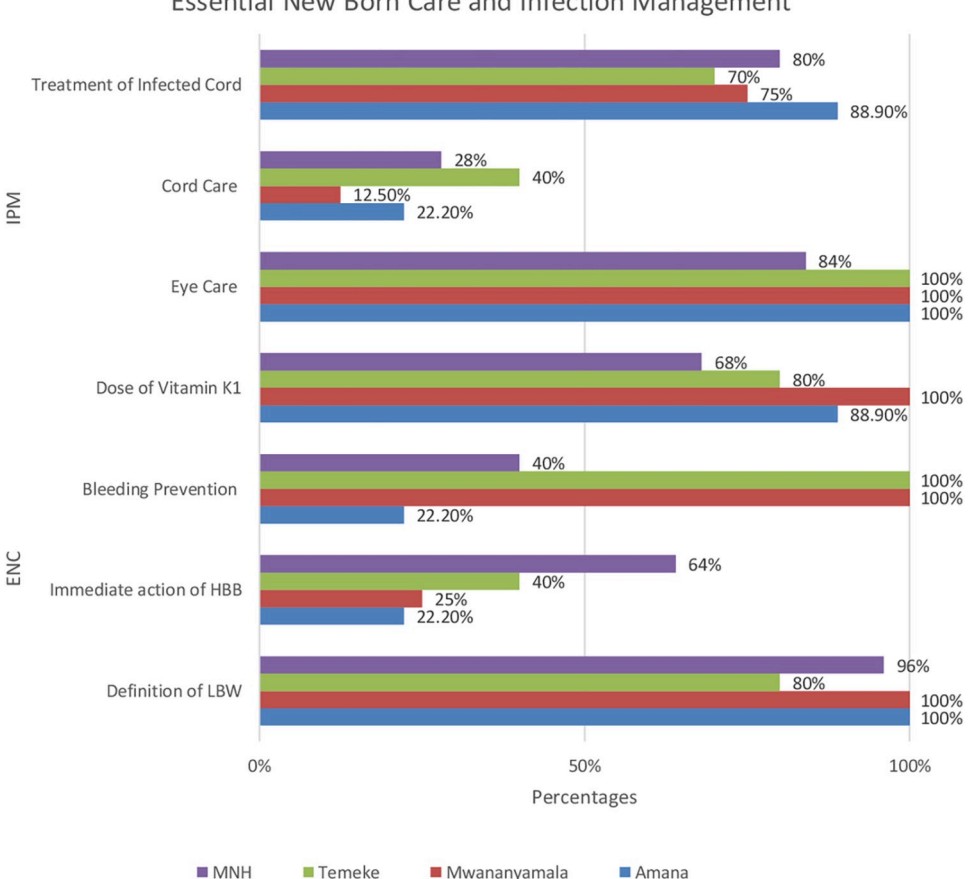

HBB -Help Baby Breath, LBW- Low Birth Weight

**Fig 2. Nurses' knowledge on Essential Newborn Care (ENC) and Infection Prevention and Management (IPM) domain by hospital.**

Most of the male and female nurses had adequate knowledge but there was no association between sex and knowledge (p = 1). Irrespective of the cadre of nursing, years of service, training attended, and level of education majority of the participants had adequate knowledge in essential newborn care but there was no association between their knowledge and the independent variable (Table 1).

As shown in Table 2 in infection management, nurses who were more than 41 years had adequate knowledge compared to other age groups with 14/15(93.3%) having adequate knowledge. Based on the multivariate analysis, there was no statistical significance in the knowledge among different age groups with a p-value of 0.689 (AOR 1.55, CI (0.1–13.31) in those 31 to 40 years and 0.763 (AOR 1.65, CI (0.06–42.89) in those who are more than 40 years. While in infection management, most of the nurses in the regional hospital had adequate knowledge whereas 18/25 (72%) had adequate knowledge from the national hospital. There was no statistical significance after multivariate analysis with a p-value of 0.309 (AOR 0.35, CI (0.04–2.61).

Most of the nurses that worked for more than 4 years had adequate knowledge in infection management, compared to 7/12(58.3%) of those who worked less than a year, and 19/23 (82.9%) of those have worked for 1 to 3 years had adequate knowledge as well. In multivariate

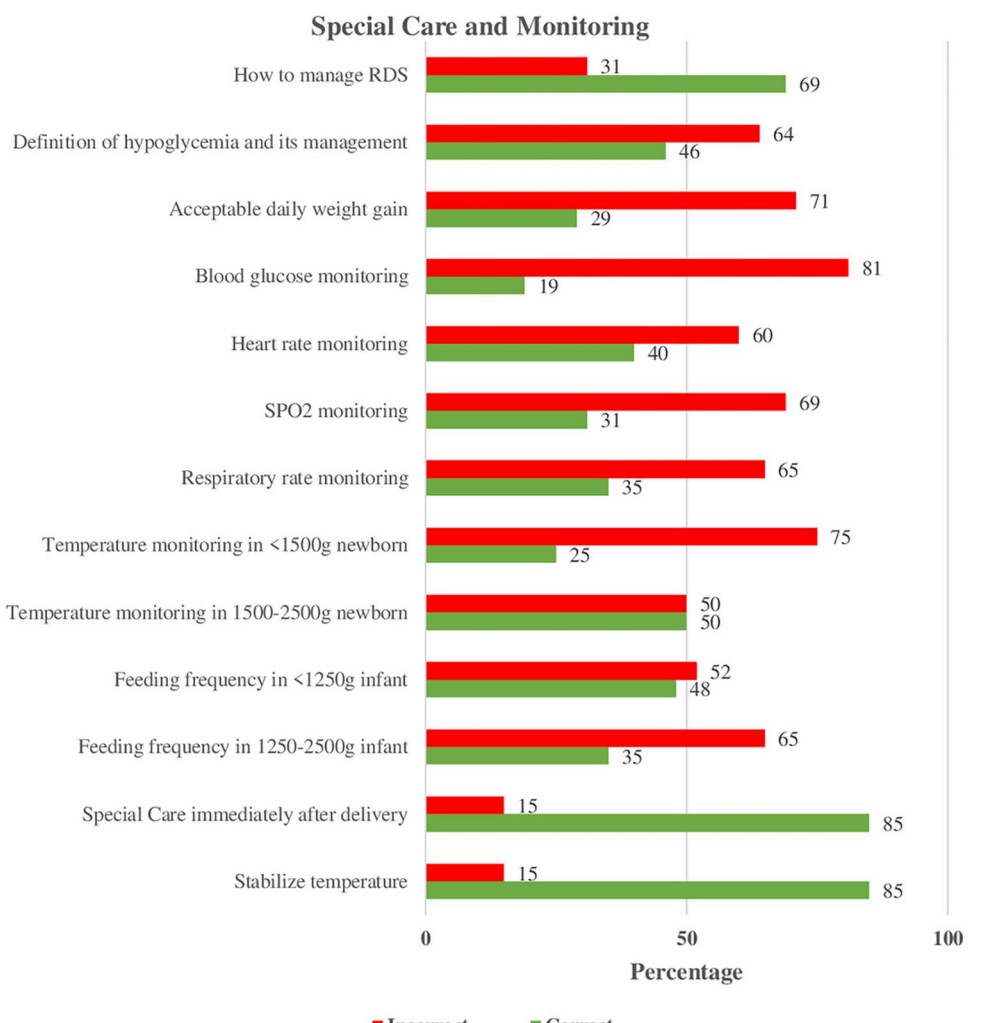

RDS - respiratory distress syndrome

**Fig 3. Nurses' knowledge on special care and monitoring domain in all hospitals.**

analysis showed a p-value of 0.592 (AOR 1.68, CI (0.25–11.25) and 0.808 (AOR 0.63, CI 0.01–24.21), there was no statistical significance between those who worked for 1 to 3 years and those who worked for more than 4 years respectively (Table 2).

The nurses from a regional hospital, 9/27(33.3%) and those from the national hospital 10/25(40%) had adequate knowledge in special care and monitoring. With the p-value of 0.618, there was no association between the hospital levels and knowledge on special care and monitoring.

In special care and monitoring, females 16/44(36.4%) and males 3/8(37.5%) had adequate knowledge. There was no association between sex and knowledge on special care (p = 1) (Table 3).

Of the enrolled nurses 7/16(43.7%) had adequate knowledge in special care and monitoring whereas 8/24(33.3%) of the assistant nursing officer and 4/12(33.3%) of the nursing officer had adequate knowledge. There was no statistical significance in knowledge among the different cadre. (p = 0.756).

**Table 1. Factors associated with nurse's knowledge on essential newborn care.**

| Variable | Categories | Inadequate knowledge | Adequate knowledge | Total | P-value |
|---|---|---|---|---|---|
| | | 3 (5.8%) | 49 (94.2%) | 52 | |
| **Age group** | 21–30 years | 2(9.5) | 19(90.5) | 21 | 0.770 |
| | 31–40 years | 1(6.2) | 15(93.8) | 16 | |
| | > 40 years | 0(0.0) | 15(100) | 15 | |
| **Hospital level** | Regional Referral hospital (RRH) | 1(3.7) | 26(96.3) | 27 | 0.603 |
| | National hospital, MNH | 2(8.0) | 23(92.0) | 25 | |
| **Gender** | Male | 0(0.0) | 8(100) | 8 | 1 |
| | Female | 3(6.8) | 41(93.2) | 44 | |
| **Cadre** | Enrolled Nurse (EN) | 0(0.00) | 16(100) | 16 | 0.220 |
| | Assistant nursing officer (ANO) | 1(4.2) | 23(95.8) | 24 | |
| | Nursing Officer (NO) | 2(16.7) | 10(83.3) | 12 | |
| **Years of service** | < 1 years | 1(8.3) | 11(91.7) | 12 | |
| | 1–3 years | 2(8.7) | 21(91.3) | 23 | 1 |
| | 4–6 years | 0(0.00) | 6(100) | 6 | |
| | > 6 years | 0(0.00) | 11(100) | 11 | |
| **Training attended** | Not attended | 2(6.9) | 27(93.1) | 29 | 1 |
| | Attended < 1 year ago | 1(8.3) | 11(91.7) | 12 | |
| | Attended > 1 year ago | 0(0.00) | 11(100) | 11 | |
| **Level of education** | Certificate | 0(0.00) | 16(100) | 16 | 0.220 |
| | Diploma | 1(4.2) | 23(95.8) | 24 | |
| | Degree | 2(16.7) | 10(83.3) | 12 | |

**Table 2. Factors associated with nurse's knowledge in infection prevention management using the binary logistic regression model.**

| | Knowledge Status | | | | | |
|---|---|---|---|---|---|---|
| Variable | Inadequate | Adequate | Bivariate analysis | | Multivariate analysis | |
| | 10 (19.2%) | 42 (80.8%) | COR (95% CI) | P value | AOR (95% CI) | P value |
| **Age group** | | | | | | |
| 21–30 years | 7(33.3) | 14(66.7) | Ref | Ref | Ref | Ref |
| 31–40 years | 2(12.5) | 14(87.5) | 3.5(0.62–19.88) | 0.158 | 1.55(0.18–13.31) | 0.689 |
| > 40 years | 1(6.7) | 14(93.3) | 7(0.75–64.61) | 0.086 | 1.65(0.06–42.89) | 0.763 |
| **Hospital level** | | | | | | |
| Regional referral hospital | 3(11.1) | 24(88.9) | Ref | Ref | Ref | Ref |
| National hospital | 7(28.0) | 18(72.0) | 0.32(0.07–1.41) | 0.134 | 0.35(0.04–2.61) | 0.309 |
| **Gender** | | | | | | |
| Male | 4(50) | 4(50) | Ref | Ref | Ref | Ref |
| Female | 6(13.6) | 38(86.4) | 6.33(1.23–32.37) | 0.027 | 3.22(0.54–19.19) | 0.198 |
| **Years of service** | | | | | | |
| < 1 years | 5(41.7) | 7(58.3) | Ref | Ref | Ref | Ref |
| 1–3 years | 4(17.4) | 19(82.6) | 3.39(0.70–16.38) | 0.128 | 1.68(0.25–11.25) | 0.592 |
| 4–6 years | 1(16.7) | 5(83.3) | 3.57(0.31–40.75) | 0.305 | 0.63(0.01–24.21) | 0.808 |
| > 6 years | 0(0.00) | 11(100) | 1 | | | |

AOR- Adjusted Odds Ratio, COR- Crudes Odds Ratio

In the special care and monitoring those aged more than 41 years 8/15 (53.3%) had adequate knowledge compared to other age groups. There was no association between the knowledge in special care and monitoring and different age groups. (p = 0.32) as summarized in Table 3.

**Table 3. Factors associated with nurse's knowledge in special care and monitoring.**

| Variable | Categories | Inadequate knowledge | Adequate knowledge | Total | P-value |
|---|---|---|---|---|---|
| | | **33 (63.5%)** | **19 (36.5%)** | **52** | |
| Age group | 21–30 years | 15(71.4) | 6(28.6) | 21 | 0.320* |
| | 31–40 years | 11(68.8) | 5(31.2) | 16 | |
| | > 40 years | 7(46.7) | 8(53.3) | 15 | |
| Hospital level | Regional Referral hospitals | 18(66.7) | 9(33.3) | 27 | 0.618* |
| | National hospital, MNH | 15(60.0) | 10(40.0) | 25 | |
| Gender | Male | 5(62.5) | 3(37.5) | 8 | 1 |
| | Female | 28(63.6) | 16(36.4) | 44 | |
| Cadre | EN | 9(56.3) | 7(43.7) | 16 | 0.756 |
| | ANO | 16(66.7) | 8(33.3) | 24 | |
| | NO | 8(66.7) | 4(33.3) | 12 | |
| Years of service | < 1 years | 9(75) | 3(25) | 12 | 0.390 |
| | 1–3 years | 16(69.6) | 7(30.4) | 23 | |
| | 4–6 years | 3(50) | 3(50) | 6 | |
| | > 6 years | 5(45.5) | 6(54.5) | 11 | |
| Training attended | None | 20(68.9) | 9(31.1) | 29 | 0.300* |
| | < 1 years | 7(58.3) | 5(41.7) | 12 | |
| | > = 1 year | 6(54.5) | 5(45.5) | 11 | |
| Level of education | Certificate | 9(56.3) | 7(43.7) | 16 | 0.760 |
| | Diploma | 16(66.7) | 8(33.3) | 20 | |
| | Degree | 8(66.7) | 4(33.3) | 12 | |

* = chi square p value

In special care and monitoring, the nurses who worked for less than a year 3/12(25%), those that worked for 1 to 3 years 7/23(30.4%), those who have worked for 4 to 6 years 3/6(50%), and those that worked for more than 6 years 6/11(54.5%) had adequate knowledge. There was no association between adequate knowledge of special care and years of experience (p = 0.39).

Those who have never attended any training, 9/29(31.1%) had adequate knowledge while among those who attended training less than a year ago, 5/12(41.7%) and those who attended more than a year ago, 5/11(45.5%) had adequate knowledge. Although there was difference in knowledge, there was no association between training attended or not attended with knowledge on special care and monitoring (p = 0.3).

Among those who held a certificate, 7/16(43.7%) had adequate knowledge in special care in monitoring while those that held a diploma, 8/24 (33.3%) had adequate knowledge and 4/12 (33.3%) among those who had a degree had adequate knowledge. There was no association between the level of education and knowledge on special care and monitoring (p = 0.76) as summarized in Table 3.

## Qualitative results

Three main themes emerged from the focused group discussion (FGD).

**General experience in acquiring knowledge.** The general nurses' experience in acquiring knowledge was based on their 1) specific knowledge needs and 2) sources they use to acquire knowledge.

**Specific knowledge needs.** When asked about what their specific knowledge needs were, two main subjects were mentioned, 1) to understand how to use the equipment 2) current

updates on the care of premature infants. The need to know how to operate equipment was common among nurses working in both the regional and national hospitals:

"...*How to educate one another on proper care of this equipment and the updates that are available on this equipment that don't reach many of the nurses...*" (P2 MNH, Upanga)

"...*knowledge that we need most is how to operate the machines (i.e. concentrator) that are brought. We are not taught how to use them. They just show us the parts. The only thing we understand is when it doesn't bubble but we don't know how to operate them...*" (P2 Temeke RRH)

The participants in both hospitals stated their need to get current updates on the care of premature infants:

"...*We understand that there are current updates in managing them. We feel we need to be informed on these current updates...*" (P4 MNH, Upanga)

"...*Maybe we should have Continuous Medical Education (CME) on helping baby breath (HBB)...*" (P5 Temeke RRH)

**Source of knowledge.**  Means through which the nurses used to acquire knowledge whenever they needed it while at work were through senior nurses, doctors, guidelines, and CME held amongst themselves.

"*.... We assist one another as nurses, we use neonatal national guideline and national infection prevention and control guideline. We do have guidelines...*" (P2 Temeke RRH)

"*...I usually ask a senior nurse who is more experienced than me to assist me first. Then I ask a doctor to assist me or when need be, we do teamwork with the nurses and doctors...*" (P5 MNH, Upanga)

"...*We usually hold CME whereby one is assigned a topic based on standard operating procedures (SOP) or guideline to teach each other in our departments...*" (P2 MNH, Upanga)

**Perceived challenges.**  Challenges started by the participants were mainly in two areas, 1) while at work and 2) while acquiring knowledge.

**Challenges perceived while working.**  The main challenges attributed to working environment included 1) inadequate staffing, 2) work overload, 3) inadequate resources, and 4) inability to use the available equipment.

Inadequate resources were mentioned as not having enough equipment and the lack of medication. These were perceived in both regional hospitals and the national hospital:

"...*For instance, when a newborn is brought in sick you will find that one bed can have two or even three or even four babies...*" (P2 Temeke RRH)

"*Another challenge is that we have few warmers...*" (P3 Temeke RRH)

"*Medication is still an issue to be dealt with by the government...*" (P2 MNH, Upanga)

Work overload was an emphasized perceived challenge that was associated with inadequate staffing. Surprisingly, this seems to be a challenge perceived mainly by nurses working at the national hospital. Some of the participants stated that:

*"...The biggest problem right now is the lack of nursing staff. The number of nurses doesn't correspond to the number of admitted premature infants, so the close monitoring and care provided isn't sufficient..."* (P2 MNH, Upanga)

*"...in these coming months of January and February we expect an increase in delivery of newborns and with the understaffing issue at hand, it becomes a big challenge ..."* (P1 MNH, Upanga)

On the issue of the inability to use the equipment while at work, this was a challenge shared from both the regional and national hospitals among the nurses:

*"... Most of the nurses don't know how to care for the equipment as needed so that becomes a challenge..."* (P2 MNH, Upanga)

*".... We are not taught on how to use them. They just show us the parts. The only thing we understand is when it doesn't bubble but we don't know how to operate them..."* (P2 Temeke RRH)

**Challenges perceived while acquiring knowledge.**   These were attributed mainly to 1) lack of support system, 2) poor selection process for attending training and 3) the inadequate number of trainings being held.

The support system was mainly referred to the government, the hospital, and fellow nurses. Both at the regional and national hospital, participants felt like the government played little role in necessitating training among the nurses:

*"One of the nurses mentioned that stakeholders (CCBRT) play a big part in helping us acquiring the knowledge but not much cooperation from the ministry..."* (P5 Temeke RRH)

*"...Mostly I have never heard of the hospital making arrangements on training to be done or maybe we are misinformed but most of the time its foreigners from other countries through CCBRT who come and teach us the updates on the management of these preterm but not the hospital..."* (P2 MNH, Upanga)

Nurses mostly learn from each other, but challenges persist for those working at the national hospital:

*"...The senior nurses are busy because they might also have twenty children to care for, resuscitation to do, and many other things..."* (P1 MNH, Upanga)

*"...It all depends on the mood of the person you are seeking assistance from. The response you get sometimes isn't good..."* (P2 MNH, Upanga)

Unfortunately, nurses both in the regional and national hospitals are faced with the challenge of attending training since most of them can't go at the same time and the number of trainings being held isn't enough to enable most of them to attend:

*"Most of the time stakeholders assist us in getting the training but it is not that frequent..."* (P5 Temeke RRH)

*"When it comes to training, not all are selected to attend but just a few of the nurses ..."* (P5 MNH, Upanga)

*". . .When it comes to deciding who goes for training, it's the task of the in-charge nurse of the ward. But the issue isn't with just the in-charge nurse, the issue is the number of nurses that could be released to go for training. . ."* (P2 MNH, Upanga)

**Solutions to the perceived challenges.**    Solutions suggested to tackle work-related challenges and those when acquiring knowledge are summarized below

**The solution to work-related challenges.**    The participants suggested that to deal with challenges while working, 1) on job training should be provided for any new nurse working in the department and 2) there should be an increase in nurse staffing to ensure proper care for the newborns:

*". . .But if given a few patients, I will be more efficient by using the knowledge I acquired from the university. . ."* (P2 MNH, Upanga)

*". . .Those who are new to the assigned ward either newly employed or just a volunteer or even a doctor, they should undergo an on-job training on how to care for the preterm. . ."* (P5 MNH, Upanga)

**The solution to improving knowledge.**    Solutions that were suggested in improving knowledge included additional of more training and increasing the number of CME being held:

*". . .More frequent training should be conducted" ". . .maybe four times a year. . ."* (P3 Temeke RRH)

*". . .At least learning should be continuous in that every nurse should get the chance to attend training so each one of us could learn. . ."* (P3 MNH, Upanga)

*". . .The weekly CME should be consistent maybe every Monday they should take place and that anyone assigned to teach a certain topic should be arranged based on their duty Rota to ensure no one is assigned a topic while they are off. . ."* (P2 MNH, Upanga)

Some of the participants advocated learning from other nurses.

*". . .Amongst the nurses at our workplaces can teach one another about something new they learned. . ."* (P1 Temeke RRH)

*". . .Maybe allowing nurses from other facilities with experience to come and teach us at our workplace. . ."* (P5 Temeke RRH)

*". . .We should ensure that they are held every week like Mondays to stimulate us. It will benefit even the newly employed nurses like how to put in an IV line and how to maintain it. . ."* (P2 MNH, Upanga)

Among the nurses at national hospital advocated for the availability of guidelines at the workplace.

*". . .That we should have a treatment guideline not only for the doctors but also the nurses need it during their duties especially the new nurses. . ."* (P4 MNH, Upanga)

## Discussion

Despite World Health Organization (WHO) strategies emphasizing on health facility capacity improvement towards investment on high quality of newborn care, challenges remain to be tackled on human resource competences [14]. In Tanzania, with most birth happening in facilities, health professionals working in the neonatal units must have specific competences to provide appropriate comprehensive newborn care [2]. Quality of health services should be effective, safe, people centered, timely, equitable, integrated, and efficient as introduced by the WHO quality toolkit [15]. Improving quality is a process which requires planning at different levels of the system [14]. While understaffing could be the main problem, ensuring competences of the few available providing care is critical.

In our study most of the nurses (94%) had adequate knowledge in essential newborn care. All the nurses aged 40 years and above had adequate knowledge on essential newborn care. Overall, in this study, there was low knowledge on step wise management of helping baby breath (HBB) algorithm, all referral hospitals scored 40% and below except for MNH (64%). Results were consistent with the study from Uganda where 21.9% of the nurses had the knowledge [10]. Evaluation of HBB in 15 regions of Tanzania showed a high drop in retention of knowledge and skills with 57% of providers who could perform well at 4–6 months after training compared to 87% immediate after training [16]. The skills attained during the training need to change the clinical practices in our routine care, surprisingly this may not be true. Can simulation training in neonatal care be a solution? [17], a low dose high frequency approach? the impact of these approach need to be documented to support what need to be changed.

In our study 55.8% of the nurses had no privilege to attend any training on premature infant care. COVID pandemic prohibiting social gatherings could be one global explanation, but qualitative findings point to lack of system for on job training at institutional and government level. Nurses have no routine schedule for training, neither mentorship plan. Gaps are identified to new less experienced staff and volunteers who need on job training and orientation before delivering care to the preterm infants. Identification of core competences and scheduling on job trainings should be mandatory and its part of quality care considering safety and effectiveness of interventions provided. According to WHO, quality care should be safe, effective and people centered. The proxy of poor quality in this study could be reflected on low level of knowledge in HBB algorithm (ranging as low as 22–64%) and special care and monitoring of preterm infants (36.5%).

During the focus group discussion, the nurses stated a poor selection to those who should attend training, there were less training held each year and that they needed an update of information on how to care for preterm infants. According to WHO's a new roadmap on human resources to ensure all newborns survive and thrive, on-the-job training has been suggested as one of the strategies for improvement to enable adequate care provided [14]. Clearly, there is a need to have on the job training guide and mentorship plans at the national, subnational/district, facility and unit level to continuously maintain competences required to care for preterm infants. Among the recommended solution in the focus group discussion, the nurses suggested to learn from each other. Perhaps the nurses who are more than 40 years can be identified as mentors, provided with skills and teach others as part of routine schedule.

In this study, most of the nurses (80.8%) had adequate knowledge on infection prevention and management compared to the study from Uganda, 13.3% [10]. Umbilical cord care by ensuring the use of a clean blade when cutting the cord as means of prevention of infection was the most poorly answered question in this domain. Whereas in the study done in Uganda this similar question had a better response with 72.1% having a correct response [10]. While the study from Ethiopia, 61% of the nurses had a correct response [18]. In a cross-sectional

study with an observational component including 6 sub-Saharan African countries including Tanzania showed that 94% of the health care workers used a clean blade to cut the cord [19]. Despite cord care having the poorest response, the practiced care is known by many health care workers. In our study the poor response might be explained by the fact that initial cord care, which was what the question was assessing, is provided by labor ward nurses and the midwife. In our study, the nurses who participated were mainly neonatal nurses, who did not practice daily on cord care.

Shockingly, overall, only 36.5% of the nurses had adequate knowledge of special care and monitoring of preterm infants. The results from our study would be speculated by the fact that there is understaffing and not having guideline for the nurses on how to conduct monitoring as reported in focus group discussion. Regular and scheduled monitoring becomes a challenge when few nurses are available to provide care. Vital signs monitoring in neonates is the most integral part of the care whereby in modern medicine, continuous vital signs monitoring is being preferred instead of intermittent monitoring [20]. It is known that trends in vital signs can predict some of the complications that the preterm newborns are at risk of such as sepsis, necrotizing enterocolitis (NEC), brain injury, broncho pulmonary dysplasia (BPD), and even mortality. This emphasizes the importance of constant monitoring of vital signs to improve care and survival for these patients.

Among the poorly answered questions was the monitoring of blood glucose with 19.2% of the nurses having correct responses. This is very crucial in the management of premature infants because impaired glucose control in very preterm newborns has been associated with an increase in morbidity, poor neurologic outcome, and even mortality [21]. Findings in our study could be due to the inability to frequently monitor random blood glucose due to inadequate supply of machines and strips which was captured in the focus group discussion, and this limits the practice of frequent monitoring.

Monitoring of temperature in those premature infants with weight less than 1500g was not well known, with only 25% of the nurses having the correct response. Lack of adequate nursing staff might contribute to lack of proper monitoring. This might also be due to the availability of continuous temperature monitoring devices for premature infants placed on a servo machine. This might not be the case for every premature infant admitted to the neonatal unit. Only the high-risk premature infants are placed under the servo machine and every preterm irrespective of being high risk or not needing monitoring. Newborns who are at risk of hypothermia are those with very low weight and those with low gestational age at the time of birth. This has a detrimental effect on their neurodevelopment or even leads to death [22]. The importance of proper temperature monitoring in these premature infants and the need to improve knowledge should be prioritized.

28.1% of the nurses knew acceptable daily weight gain of the premature infant despite being a daily routine practice done in the neonatal unit. Weight monitoring is crucial since it is an indicator of the energy intake and expenditure (internal heat loss through basal metabolism and physical activity) and enables one to decide on how to intervene when there isn't adequate weight gain i.e., the use of parenteral nutrition. Poor growth during hospitalization in the neonatal unit has been linked to poor neurodevelopmental outcomes [23]. There is a need to emphasize on the proper daily weight monitoring to ensure proper decision and action is put in place.

Furthermore, this study found that nurses who were older than 41 years of age were two times more likely to have adequate knowledge on infection prevention and management compared to those who were less than 30 years. This was similar to a study done in Mosul whereby nurses aged 40–49 years had more knowledge on general care of premature infants compared to the younger nurses [24]. Nurses that worked for 1–3 years were 2 times more likely to have

adequate knowledge in infection prevention and management compared to those with more years of experience. This could also be explained by the fact that the nurses with less years of experience in neonatal ward were updated on current care practices since they were recently from school. Females' nurses were 3.2 times more likely to have adequate knowledge in infection prevention and management compared to male nurses. This could be explained by the high numbers of female nursing staff compared to the males. During the focus group discussion, nurses in our study stated that their sources of knowledge is from senior nurses. On the contrary, nurses at the national level were less likely to have adequate knowledge on infection management compared to those in regional referral hospitals. Building a mentorship plan between hospitals should be considered a priority. For example, where competence and best practices are identified, that hospital or person can be targeted to mentor and share experience with others. In our study peer to peer learning was among the suggested comfortable way of learning.

Our study had limited sample size due to low number of neonatal care units and hence nurses recruited thus, the generalizability of the data becomes a challenge. Only assessment of knowledge was conducted using a structured questionnaire, but we did not observe the skills. If nurses have inadequate knowledge without assessing their practice, we cannot ascertain their knowledge and validate their practice. A larger observational study is recommended to ascertain adequate care provided rather than making this assumption with only assessment of knowledge. Despite the limitations, the mixed method approach allowed us to understand the gap on nurses' knowledge, challenges, and solutions to support an urgent intervention to quality care of preterm infants in Tanzania. The findings demonstrate an urgent need of instilling knowledge, skills and competences among nurses providing preterm care in our hospitals. National operational guide for integrated competence based clinical mentorship for neonatal services could be among the suggested policy required. Other suggestions include i) Structured continuous medical education (CME) ii) On the job competence need assessment and training plans iii) Government and hospital commitment to ensure effective implementation and quality care in the neonatal units.

## Supporting information

**S1 File. Self-administered questionnaire English version.**
(DOCX)

**S2 File. Interview guide questions.**
(DOCX)

**S3 File. COREQ (COnsolidated criteria for REporting Qualitative research) checklist.**
(DOCX)

## Acknowledgments

The authors would like to thank all nurses for participating in this study. In addition, we acknowledge officials of hospitals at different levels for their support. A special thanks to the Department of Paediatrics and Child Health, Muhimbili university of Health and Allied Sciences, (MUHAS) for all the supervision support to implement this research. Lastly we would like to thank the Government of Tanzania through the Ministry of Health, Community Development, Gender, Elderly and Children for their support.

## Author Contributions

**Conceptualization:** Mwajuma Mwikali, Nahya Salim, Isabella Sylvester, Emmanuel Munubhi.

**Data curation:** Mwajuma Mwikali, Nahya Salim, Isabella Sylvester, Emmanuel Munubhi.

**Formal analysis:** Mwajuma Mwikali, Nahya Salim, Isabella Sylvester, Emmanuel Munubhi.

**Funding acquisition:** Mwajuma Mwikali.

**Investigation:** Mwajuma Mwikali.

**Methodology:** Mwajuma Mwikali, Nahya Salim, Isabella Sylvester, Emmanuel Munubhi.

**Project administration:** Mwajuma Mwikali, Nahya Salim.

**Resources:** Mwajuma Mwikali.

**Supervision:** Nahya Salim, Emmanuel Munubhi.

**Validation:** Nahya Salim.

**Visualization:** Nahya Salim, Isabella Sylvester.

**Writing – original draft:** Mwajuma Mwikali.

**Writing – review & editing:** Mwajuma Mwikali, Nahya Salim, Emmanuel Munubhi.

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
