## [Decision Letter · Decision Letter 0]

21 Sep 2022

PONE-D-21-38982Nurses’ knowledge, challenges and solutions on the care of premature infants: a mixed method study in the referral and tertiary hospitals in Dar es salaamPLOS ONE

Dear Dr.Mwajuma Mwikali Mwamtenda,

Thank you for submitting your manuscript to PLOS ONE. After careful consideration, we feel that it has merit but does not fully meet PLOS ONE’s publication criteria as it currently stands. Therefore, we invite you to submit a revised version of the manuscript that addresses the points raised during the review process.

ACADEMIC EDITOR: The topic is quite relevant but the manuscript needs to be extensively revised as sughgested by two of the reviewers who have pointed out key issues in the manuscript that needs to be adressed.Prominet amongst them is the need to provide clearity on how the sample size was determined as well as a clear presentaion of the statistical analysis in the result section.The discusion should be in-depth and also harmonize the relationship between the quantitative study and qualitative study fidnings.Other specific comments are noted below for your kind attention.

Introduction

This section has been written as independent sentences without considering bringing these sentences with similar context as paragraphs. For example, lines 1-6 should be under one paragraph, lines 7-15 should form another paragraph.

This section is significantly deficient as it has not highlighted whether there had been previous studies on this subject in Tanzania and what justifies the need for the current study.

Methodology

Page 11, line 6: and association factors

Comment: Please change to associated factors

Page 11,lines 13-18: This study was conducted in the three RRHs and one of the tertiary branches, MNH- Upanga. MNH-Upanga alone accounts for approximately 369 preterm deliveries per month not accounting for the referral ins (19) with bed capacity of approximately 100 in the specific premature ward. In the regional referral hospitals, preterm deliveries can be approximately 42 per month in all the hospitals (20) with a bed capacity of approximately 15 to 20 which is for both term and preterm newborns

Comment: The reason for the (19) and (20) reflected in the sentences are not clear.

Consider replacing the phrase “not accounting for referral ins” to “excluding the referral ins”

Sample size calculation

The authors should clearly and explicitly describe how the sample size of 5o was derived, please also provide references for the formula and 69% proportion that was quoted. This is very important for the interpretation of the results of the study.

Pilot study: Where was the pilot study done, how many patients were involved in the pilot study? What were the observed issues that were corrected?

Sampling: How did the authors decide on the number of nurses to be interviewed in the participating hospital? This should be highlighted.

Qualitative study design

Selection of nurses for qualitative study: From the regional hospital, the hospital with the most nurses with inadequate knowledge in special care and monitoring was selected.

Comment: Please explain the rationale for this step

Additionally, this section has not been reported according to The Consolidated Criteria For Reporting Qualitative Studies (COREQ). This section should be systematically presented in accordance with COREQ, and the checklist must be attached as supplementary file during resubmission.

Results

This section, particularly the qualitative component, has not been well presented. There is need for a univariate analysis before multiple regression analysis in the quantitative arm of the presentation of findings while the presentation of findings following the themes that emanated from the FDGs are quite scanty for comprehension by readers.

Discussion

Overall, this section has been poorly written without considering the essence of the mixed methods study design. There is no effort at triangulation of results and there has not been any in-depth discussion of the study’s findings. In the quantitative arm, the studies used to compare findings were limited. For example, knowledge on essential newborn care (ENC): There is a fault in comparing two studies with different study populations. While the current study was conducted amongst nurses involved in the management of premature babies, the study in Uganda was carried out amongst outpatient nurses. Therefore, it is not surprising to have a higher knowledge in the former group. Please rewrite this section.

Study limitations: Please write as prose and delete item 3.

Conclusion: Please write as prose.

Introduction

This section has been written as independent sentences without considering bringing these sentences with similar context as paragraphs. For example, lines 1-6 should be under one paragraph, lines 7-15 should form another paragraph.

This section is significantly deficient as it has not highlighted whether there had been previous studies on this subject in Tanzania and what justifies the need for the current study.

Methodology

Page 11, line 6: and association factors

Comment: Please change to associated factors

Page 11,lines 13-18: This study was conducted in the three RRHs and one of the tertiary branches, MNH- Upanga. MNH-Upanga alone accounts for approximately 369 preterm deliveries per month not accounting for the referral ins (19) with bed capacity of approximately 100 in the specific premature ward. In the regional referral hospitals, preterm deliveries can be approximately 42 per month in all the hospitals (20) with a bed capacity of approximately 15 to 20 which is for both term and preterm newborns

Comment: The reason for the (19) and (20) reflected in the sentences are not clear.

Consider replacing the phrase “not accounting for referral ins” to “excluding the referral ins”

Sample size calculation

The authors should clearly and explicitly describe how the sample size of 5o was derived, please also provide references for the formula and 69% proportion that was quoted. This is very important for the interpretation of the results of the study.

Pilot study: Where was the pilot study done, how many patients were involved in the pilot study? What were the observed issues that were corrected?

Sampling: How did the authors decide on the number of nurses to be interviewed in the participating hospital? This should be highlighted.

Qualitative study design

Selection of nurses for qualitative study: From the regional hospital, the hospital with the most nurses with inadequate knowledge in special care and monitoring was selected.

Comment: Please explain the rationale for this step

Additionally, this section has not been reported according to The Consolidated Criteria For Reporting Qualitative Studies (COREQ). This section should be systematically presented in accordance with COREQ, and the checklist must be attached as supplementary file during resubmission.

Results

This section, particularly the qualitative component, has not been well presented. There is need for a univariate analysis before multiple regression analysis in the quantitative arm of the presentation of findings while the presentation of findings following the themes that emanated from the FDGs are quite scanty for comprehension by readers.

Discussion

Overall, this section has been poorly written without considering the essence of the mixed methods study design. There is no effort at triangulation of results and there has not been any in-depth discussion of the study’s findings. In the quantitative arm, the studies used to compare findings were limited. For example, knowledge on essential newborn care (ENC): There is a fault in comparing two studies with different study populations. While the current study was conducted amongst nurses involved in the management of premature babies, the study in Uganda was carried out amongst outpatient nurses. Therefore, it is not surprising to have a higher knowledge in the former group. Please rewrite this section.

Study limitations: Please write as prose and delete item 3.

Conclusion: Please write as prose.

We look forward to receiving your revised manuscript.

Kind regards,

Godwin Otuodichinma Akaba, MBBS,MSc,MPH,FWACS

Academic Editor

PLOS ONE

Journal Requirements:

Additional Editor Comments (if provided):

Introduction

This section has been written as independent sentences without considering bringing these sentences with similar context as paragraphs. For example, lines 1-6 should be under one paragraph, lines 7-15 should form another paragraph.

This section is significantly deficient as it has not highlighted whether there had been previous studies on this subject in Tanzania and what justifies the need for the current study.

Methodology

Page 11, line 6: and association factors

Comment: Please change to associated factors

Page 11,lines 13-18: This study was conducted in the three RRHs and one of the tertiary branches, MNH- Upanga. MNH-Upanga alone accounts for approximately 369 preterm deliveries per month not accounting for the referral ins (19) with bed capacity of approximately 100 in the specific premature ward. In the regional referral hospitals, preterm deliveries can be approximately 42 per month in all the hospitals (20) with a bed capacity of approximately 15 to 20 which is for both term and preterm newborns

Comment: The reason for the (19) and (20) reflected in the sentences are not clear.

Consider replacing the phrase “not accounting for referral ins” to “excluding the referral ins”

Sample size calculation

The authors should clearly and explicitly describe how the sample size of 5o was derived, please also provide references for the formula and 69% proportion that was quoted. This is very important for the interpretation of the results of the study.

Pilot study: Where was the pilot study done, how many patients were involved in the pilot study? What were the observed issues that were corrected?

Sampling: How did the authors decide on the number of nurses to be interviewed in the participating hospital? This should be highlighted.

Qualitative study design

Selection of nurses for qualitative study: From the regional hospital, the hospital with the most nurses with inadequate knowledge in special care and monitoring was selected.

Comment: Please explain the rationale for this step

Additionally, this section has not been reported according to The Consolidated Criteria For Reporting Qualitative Studies (COREQ). This section should be systematically presented in accordance with COREQ, and the checklist must be attached as supplementary file during resubmission.

Results

This section, particularly the qualitative component, has not been well presented. There is need for a univariate analysis before multiple regression analysis in the quantitative arm of the presentation of findings while the presentation of findings following the themes that emanated from the FDGs are quite scanty for comprehension by readers.

Discussion

Overall, this section has been poorly written without considering the essence of the mixed methods study design. There is no effort at triangulation of results and there has not been any in-depth discussion of the study’s findings. In the quantitative arm, the studies used to compare findings were limited. For example, knowledge on essential newborn care (ENC): There is a fault in comparing two studies with different study populations. While the current study was conducted amongst nurses involved in the management of premature babies, the study in Uganda was carried out amongst outpatient nurses. Therefore, it is not surprising to have a higher knowledge in the former group. Please rewrite this section.

Study limitations: Please write as prose and delete item 3.

Conclusion: Please write as prose.

Reviewers' comments:

Reviewer's Responses to Questions

**Comments to the Author**

1. Is the manuscript technically sound, and do the data support the conclusions?

Reviewer #1: Yes

Reviewer #2: Partly

Reviewer #3: Partly

2. Has the statistical analysis been performed appropriately and rigorously? 

Reviewer #1: Yes

Reviewer #2: No

Reviewer #3: Yes

3. Have the authors made all data underlying the findings in their manuscript fully available?

Reviewer #1: Yes

Reviewer #2: Yes

Reviewer #3: Yes

4. Is the manuscript presented in an intelligible fashion and written in standard English?

Reviewer #1: Yes

Reviewer #2: No

Reviewer #3: Yes

5. Review Comments to the Author

Reviewer #1: It is preferable to know more about the setting that the study was conducted in and definitions of prematurity.

The questionnaire is very limited. For example, there is much more to Infection control than cord care.

I do not feel that it adds much to the discussion to compare the results from this study to similar ones conducted in other countries.

Reviewer #2: This manuscript id addressing a very topical and an important subject matter. However, the following should be clarified.

1. The use of symbols is not ideal in writing.

2. In the methodology, the sample size calculation is not clearly stated. The assumptions made in thew calculation of sample size were not clarified. The research assistants were not clearly defined. Are they nurses in the hospitals used for the study or not? The statistical analysis was not explained detailed. How the adjusted odds ratio were generated was not clear. There ought to be a bivariate analysis of the different domains before a multivariate analysis. this was not the case in this manuscript.

3. Table 1 is very clumsy due to the observation stated in 2 above.

4. In the discussion, there was no triangulation of the quantitative and qualitative study. It thus appear they were different studies. Ideally there should be a link between both in the discussion. The discussion should be re-written in a more robust manner.

Reviewer #3: This article addresses an important subject of improving the survival of premature neonates, through improving the care they receive while in hospital. The authors have presented an article that has potential but will require several modifications.

Please see all comments in the uploaded file. Aspect that are not well written and require correction are highlighted.

Consider the suggested title modification for clarity as follows 'Nurses' knowledge, perceived challenges and recommended solutions regarding premature infant care; a mixed methods study in hospitals in Dar es salaam'

6. PLOS authors have the option to publish the peer review history of their article (what does this mean?). If published, this will include your full peer review and any attached files.

Reviewer #1: No

Reviewer #2: No

Reviewer #3: **Yes: **Oluchi Joan Kanma-Okafor

---

## [Decision Letter · Decision Letter 1]

18 Jan 2023

nurses' knowledge, perceived challenges and recommended solutions regarding premature infant care; a mixed method study in the referral and tertiary hospitals in Dar es salaam

PONE-D-21-38982R1

Dear Dr. Mwajuma Mwikali Mwamtenda,

We’re pleased to inform you that your manuscript has been judged scientifically suitable for publication and will be formally accepted for publication once it meets all outstanding technical requirements.

Kind regards,

Hamufare Dumisani Dumisani Mugauri, Ph.D. Public Health

Academic Editor

PLOS ONE

Additional Editor Comments (optional):

The manuscript has gone through a great deal of modifications, from the initial submission. It is now suitable for publication, with the minor comments

Reviewers' comments:

Reviewer's Responses to Questions

**Comments to the Author**

1. If the authors have adequately addressed your comments raised in a previous round of review and you feel that this manuscript is now acceptable for publication, you may indicate that here to bypass the “Comments to the Author” section, enter your conflict of interest statement in the “Confidential to Editor” section, and submit your "Accept" recommendation.

Reviewer #1: (No Response)

Reviewer #2: (No Response)

Reviewer #3: All comments have been addressed

2. Is the manuscript technically sound, and do the data support the conclusions?

Reviewer #1: Partly

Reviewer #2: Yes

Reviewer #3: Yes

3. Has the statistical analysis been performed appropriately and rigorously? 

Reviewer #1: No

Reviewer #2: Yes

Reviewer #3: Yes

4. Have the authors made all data underlying the findings in their manuscript fully available?

Reviewer #1: Yes

Reviewer #2: Yes

Reviewer #3: (No Response)

5. Is the manuscript presented in an intelligible fashion and written in standard English?

Reviewer #1: Yes

Reviewer #2: Yes

Reviewer #3: No

6. Review Comments to the Author

Reviewer #1: The study presented remains very weak due to the small sample size which was not further explained in the revision. In addition, the items in each category of pre-term newborn care remain inadequate and poorly described for proper care of the infants. The manuscript still contains references to studies in other countries which are not really comparable.

Reviewer #2: The authors should make the following corrections:

1. They should be consistent with three decimal places when writing p values in tables 1 and 3.

2. In line 438: 'tackles" should read "tackled".

3. Line 444-446: There is no need to repeat the objectives of the study which has been captured in lines 90-92 in the discussion section.

4. Lines 490 and 547: "didn't" should read "did not"

5. Line 520: Be specific about the proportion instead of stating "about 28.!%.

6. Line 548: "can't" should read "cannot"

Reviewer #3: Dear author,

The paper has undergone a great deal of modification and is currently in a good shape for publication, aside from a few errors in sentence structuring. I also do not think that you addressed all my past comments. Please attend to this by doing a proofread for identifying areas where the sentence flow is less than fluid. Thanks.

7. PLOS authors have the option to publish the peer review history of their article (what does this mean?). If published, this will include your full peer review and any attached files.

Reviewer #1: No

Reviewer #2: No

Reviewer #3: **Yes: **Oluchi Joan Kanma-Okafor

---

## [Editor Report · Acceptance letter]

13 Feb 2023

PONE-D-21-38982R1 

Nurses’ knowledge, perceived challenges, and recommended solutions regarding premature infant care: a mixed method study in the referral and tertiary hospitals in Dar es salaam, Tanzania. 

Dear Dr. Mwamtenda:

I'm pleased to inform you that your manuscript has been deemed suitable for publication in PLOS ONE. Congratulations! Your manuscript is now with our production department. 

Kind regards, 

on behalf of

Mr Hamufare Dumisani Dumisani Mugauri 

Academic Editor

PLOS ONE